# Advancing patient care: Machine learning models for predicting grade 3+ toxicities in gynecologic cancer patients treated with HDR brachytherapy

Andres Portocarrero-Bonifaz[1,2]*, Salman Syed[1], Maxwell Kassel[1], Grant W. McKenzie[1], Vishwa M. Shah[3], Bryce M. Forry[1], Jeremy T. Gaskins[4], Keith T. Sowards[1], Thulasi Babitha Avula[1], Adrianna Masters[1], Jose G. Schneider[5], Scott R. Silva[1]

1 Department of Radiation Oncology, Brown Cancer Center, University of Louisville School of Medicine, Louisville, Kentucky, United States of America, 2 Department of Radiation Oncology, Mayo Clinic, Jacksonville, Florida, United States of America, 3 Division of Gynecologic Oncology, Department of Gynecology and Obstetrics, Loma Linda University Medical Center, Loma Linda, California, United States of America, 4 Department of Bioinformatics and Biostatistics, University of Louisville School of Public Health and Information Sciences, Louisville, Kentucky, United States of America, 5 Department of Radiation Oncology, Vanderbilt University Medical Center, Nashville, Tennessee, United States of America

* aportocarrerob@pucp.edu.pe

## Abstract

### Background

Gynecological cancers are among the most prevalent cancers in women worldwide. Brachytherapy, often used as a boost to external beam radiotherapy, is integral to treatment. Advances in computation, algorithms, and data availability have popularized the use of machine learning to predict patient outcomes. Recent studies have applied models such as logistic regression, support vector machines, and deep learning networks to predict specific toxicities in patients who have undergone brachytherapy.

### Objective

To develop and compare machine learning models for predicting grade 3 or higher toxicities in gynecological cancer patients treated with high dose rate (HDR) brachytherapy, aiming to contribute to personalized radiation treatments.

### Methods

A retrospective analysis was performed on gynecological cancer patients who underwent HDR brachytherapy with Syed-Neblett or Tandem and Ovoid applicators from 2009 to 2023. After applying exclusion criteria, 233 patients were included in the analysis. Dosimetric variables for the high-risk clinical target volume (HR-CTV) and organs at risk, along with tumor, patient, and toxicity data, were collected and compared between groups with and without grade 3 or higher toxicities using statistical

**Data availability statement:** The Python code and study database have been made available to the reader. This data is available at: https://github.com/AndresPB95/ML-Model-Gynecological-HDR-G3Plus-Toxicities.

**Funding:** The author(s) received no specific funding for this work.

**Competing interests:** The authors have declared that no competing interests exist.

tests. Seven supervised classification machine learning models (Logistic Regression, Random Forest, K-Nearest Neighbors, Support Vector Machines, Gaussian Naive Bayes, Multi-Layer Perceptron Neural Networks, and XGBoost) were constructed and evaluated. The training process involved sequential feature selection (SFS) when appropriate, followed by hyperparameter tuning. Final model performance was characterized using a 25% withheld test dataset.

### Results

The top three ranking models were Support Vector Machines, Random Forest, and Logistic Regression, with F1 testing scores of 0.63, 0.57, and 0.52; normMCC testing scores of 0.75, 0.77, and 0.71; and accuracy testing scores of 0.80, 0.85, and 0.81, respectively. The SFS algorithm selected 10 features for the highest-ranking model. In traditional statistical analysis, HR-CTV volume, Charlson Comorbidity Index, Length of Follow-Up, and D2cc - Rectum differed significantly between groups with and without grade 3 or higher toxicities.

### Conclusions

Machine learning models were developed to predict grade 3 or higher toxicities, achieving satisfactory performance. Machine learning presents a novel solution to creating multivariable models for personalized radiation therapy.

### Introduction

Gynecological cancers rank among the most diagnosed malignancies affecting women on a global scale [1]. In the United States of America, it is estimated that there were 116,930 new cases and 36,250 deaths in 2024 attributable to gynecologic malignancies [2]. Treatments for gynecologic cancers include surgery, chemotherapy, and/or radiotherapy [3]. Brachytherapy is necessary in the management of locally advanced cervical cancer, since patients who do not receive brachytherapy following concurrent external beam radiation therapy (EBRT) and chemotherapy have significantly worse overall survival [4]. Colson-Fearon et al. reported that the 4-year overall survival in locally advanced cervical cancer patients treated with brachytherapy versus without brachytherapy is 67.7% versus 45.7%, respectively [5]. With 3-dimensional magnetic resonance image-guided brachytherapy for cervical cancer, the 5-year local control is 92% [6].

Possible side effects in patients who have undergone radiation treatment for gynecological cancers with a brachytherapy component include gastrointestinal (GI), genitourinary (GU), and vaginal (VAG) toxicities. According to the Common Terminology Criteria for Adverse Events (CTCAE) version 5.0, these toxicities are graded from 1 (mild) to 5 (death) based on their severity and impact on daily activities. Common GI toxicities include diarrhea, proctitis, and rectal bleeding. GU toxicities often present as urinary frequency, incontinence, or cystitis. Vaginal toxicities such as

stenosis, mucositis, and fistulas may also occur. Follow-up data from the EMBRACE-I trial reported that at three years, the incidence of severe (grade 3 or higher) GI, GU, and vaginal toxicities were 7% (5.6–8.8%), 6.1% (4.8–7.7%), and 3.6% (2.7–5.0%), respectively. The overall incidence of adverse events, including non-GI, GU, and VAG toxicities, was 21.5% at three years and 26.6% at five years [7].

Machine learning (ML) has been defined as an optimization problem to find the most suitable predictive model for new data based on an existing dataset obtained from a similar context [8]. The recent rise in popularity of ML has been due to the development of new algorithms, theory, data availability, and improvements in low-cost computation [9]. For many problems, ML has shown to have better overall predictive metrics than conventional statistical models (CSM) [10–12]. In the treatment of cervical cancer with HDR brachytherapy, ML has proven to be applicable to several aspects. Among others, tree-based models have been found to provide accurate classification performance for optimal applicator selection [13], and deep-learning methods show promise in improvement of treatment planning, such as segmentation, reconstruction, plan optimization, dose calculation, and other aspects related to treatment outcomes [14].

ML is a bottom-up approach that has the advantages of being data-driven, of not requiring strict a-priori assumptions about the forms of the relationships between variables and outcomes, and of accounting for complex interactions among input features [15]. In contrast, CSMs can be viewed as top-down approaches, and their main advantages are their interpretability due to usually focusing on the parsimonious relationships between input and response, the low computational resources needed to fit the models, and being less susceptible to overfitting with large datasets [16,17].

Binary classification, in which the ML model predicts an output that is either one of two possible classes, is one of the most common tasks that can be solved with supervised machine learning [18]. For this problem, a model is trained with data that contains both features and the response labels, and the algorithm compares the actual and predicted results using an appropriate assessment metric [19]. This study aims to build and compare some of the more common binary classification machine learning models in the context of predicting if a patient will develop grade 3 or higher toxicities (Output: Yes/No) in gynecologic cancer patients treated with EBRT and brachytherapy. By accurately forecasting severe toxicities based on clinical, demographic, and treatment-related features, we aim to enhance personalized patient care by enabling clinicians to identify high-risk patients before treatment initiation. This predictive capability allows for proactive adjustments to treatment plans and implementation of preventive measures regarding potential side effects. This study forms part of a larger research effort aimed at improving treatment outcomes and quality of life (QoL) for patients by integrating machine learning models into the clinical decision-making process.

## Methods

### Data collection

A comprehensive retrospective analysis was conducted, encompassing a total of 233 patients who had undergone high dose rate (HDR) brachytherapy with Syed-Neblett or Fletcher-Suit-Delclos Tandem and Ovoid (T&O) applicators for treatment of gynecological cancer (cervix, endometrium, vagina, or vulva) at a single institution spanning the period from 2009 to 2023. As healthcare professionals directly involved in patient care, the authors had access to identifying patient information throughout the study. This data was accessed for research purposes from March 1st, 2022, to March 1st, 2024. Demographic details, tumor characteristics, treatment variables, dosimetric information (including if the patient received an EBRT boost), and occurrences of gastrointestinal (GI), genitourinary (GU), and vaginal (VAG) toxicities during and post-treatment were gathered. The exclusion criteria included the following: patients with a prior brachytherapy history, those treated with more than a single type of brachytherapy applicator, conflicting dosimetric data found in records, concurrent external beam radiotherapy for a distinct proximal disease site, or a combination of low dose rate (LDR) and HDR treatments. Toxicities were classified according to the Common Terminology Criteria for Adverse Events (CTCAE) version 5.0 [20], and the integrity of the database was reviewed three times by both a physician and a medical physicist

to ensure its accuracy and reliability. For treatment planning, the dosimetry goals as detailed in the EMBRACE trials and ASTRO Clinical Practice Guideline were followed [6,21]. All patients received EBRT and brachytherapy. The process used to calculate the total EQD2 dose has been described in detail in a previous work, and follows the procedure suggested by ICRU 89 [22,23]. This study was approved by our institutional review board (IRB 22.0117).

## Statistical analysis

Preliminary dataset exploration was done by comparing between patients that developed no higher than a grade 2 toxicity and those that developed grade 3 or higher toxicities at any point in time after EBRT initiation; continuous variables were reported as means and standard deviations and compared with 2-sample t-tests. Categorical variables were listed as counts and percentages and compared with the Fisher exact test. Non-normal continuous variables were reported at median and interquartile range (IQR) and compared with the non-parametric Mann-Whitney test; a p-value of 0.05 or lower was considered to be statistically significant. Kaplan-Meier curves for disease free survival and local control were created to characterize the cohort. This analysis was performed using R statistical software version 4.3.2. [24].

## Data preprocessing

The machine learning analysis was done using Python 3 and Jupyter Notebook (IPython kernel). Various code libraries (collections of pre-written functions and classes), including Scikit-learn v1.3.2 [25], were used for their straightforward integration and reproducibility; care was taken to ensure compatibility and the use of the appropriate library versions. Charlson Comorbidity Index was categorized into approximate quartiles ("Low" (0–2), "Medium" (3), "High" (4–5), or "Very High" (> 5)), and KPS was assigned categories according to clinical interpretation: "Bad" (50–70), "Normal" (80), or "Good" (90–100). Data pre-processing involved four steps: A) Encoding, B) Imputation, C) Class Balancing and D) Normalization.

For data encoding, categorical and ordinal variables were assigned to numeric labels. The data then underwent a stratified split based on the target, resulting in two groups with an equivalent proportion of toxicity events: 75% for training (n = 174) and 25% for testing (n = 59).

Imputation of missing feature values was done via the K-nearest neighbors (KNN) algorithm due to its simplicity, low computational cost and its better overall performance over other data imputation methods such as Multilayer Perceptron (MLP) or mean/mode imputation; this approach aligns with the recommendations of Garcia-Laencina et al [26]. Imputation was essential to allow consistent, unbiased comparison across multiple ML algorithms, including those which inherently cannot appropriately function with missing values, and to fully utilize our limited dataset without discarding cases. The missing data points primarily result from unobtainable patient records or measurements not being taken, suggesting the missingness is Missing at Random (MAR). KNN imputation is appropriate for MAR imputation when there is only a moderate amount of missingness (15%-30% missing), as in our setting, as it matches an individual with missing data to a similar patient based on the observed data; the advantages of KNN may be less pronounced with lower levels of missing data, as suggested by Beretta et al [27]. For categorical and ordinal variables, a K-nearest neighbors imputer was employed utilizing the single nearest neighbor to guarantee imputation to a single class for that feature. For the numerical continuous features, the KNN imputer was used with K = 5 neighbors, and the missing features were imputed by the average. This parameter was chosen after extensive experimentation. These imputers identify their nearest neighbors by calculating the Euclidean Distance between data points (not including the missing data). They were fitted using the training data only, and their algorithms applied to both the training and testing data [28].

The defined positive class of Grade 3 or higher toxicity was observed in a minority of patients (24%, 56/233), leading to an imbalanced dataset. To address this imbalance, the class-balancing algorithm SVM-SMOTE [29] was used only during model training. (Preliminary analyses suggested this algorithm had better performance than alternative balancing algorithms such as SMOTE [30] and ADASYN [31]). Out of the initial 174 samples from the training data, an additional 90 synthetic positive cases were generated for a total of 264 samples (132 positive, 132 negative).

The final pre-processing step included the normalization/standardization of values. After experimentation, the Standard Scaler was used for continuous numerical variables. For categorical and ordinal variables, both Target Encoding and One Hot Encoding were initially considered. Due to our sample size, the former was ultimately favored due to it not increasing the number of features, given how an increased features-to-samples ratio has been associated with overfitting the data [32]. Other normalization/standardization methods such as MinMax Scaler and the Robust Scaler were also explored but not reported in this work due to obtaining worse results. The fitted Scalers and the Target Encoding objects were stored into a Joblib file and then employed in the testing data.

Investigation into collinearity between input features was also performed using Pearson's correlation coefficient. The final model eliminated one of the pairs of collinear features with values greater than 0.80 correlation. Other thresholds such as 0.7 and 0.95 were also analyzed but yielded worse results. When dose metrics were collinear, D2cc and D90 were given priority to remain in the final model due to being the most widely used clinical values [21].

## Evaluation of machine learning models

There are multiple performance metrics that can be used to assess a model performance on predicting new data. In this study, the Accuracy, Precision, Recall, F1 score, Matthews Correlation Coefficient (MCC), the area under the curve of a receiver operating characteristic curve (AUC-ROC), and the area under the curve of a precision-recall curve (AUC-PR) were used; the first four metrics are defined using the number of True Positives (TP), True Negatives (TN), False Positives (FP) and False Negatives (FN) as follows:

$$Accuracy = \frac{TP + TN}{TP + FP + TN + FN} \tag{1}$$

$$Precision = \frac{TP}{TP + FP} \tag{2}$$

$$Recall = \frac{TP}{TP + FN} \tag{3}$$

$$F1\ Score = \frac{2 \cdot Precision \cdot Recall}{Precision + Recall} \tag{4}$$

$$MCC = \frac{(TP \cdot TN) - (FP \cdot FN)}{\sqrt{(TP + FP) \cdot (TP + FN) \cdot (TN + FP) \cdot (TN + FN)}} \tag{5}$$

In this context, Positive/Negative represents whether the ML model predicts a toxicity event, and True/False represents whether the ML prediction agrees/disagrees with the patient record. Accuracy as shown in formula (1) is the ratio of correctly predicted instances over the total number of patients. Precision, which is represented by formula (2), is the ratio of correctly predicted positive observations to the total number of observations predicted to be positive. Recall, also known as Sensitivity, is the ratio of correct predictions among patients with toxicities as shown in formula (3); the F1 score, as defined in formula (4), is equivalent to the harmonic mean of precision and recall [33]; MCC, or its normalized version (normMCC) [34], is a balanced measure that considers all four basic metrics (TP, FP, TN, FN) as shown in formula (5). Additional metrics such as the AUC-ROC and AUC-PR evaluate the overall performance of the model by considering performance across all possible decision thresholds of the model [35]. In this work, the reported F1, recall, and precision scores are calculated for the positive class (patients that present a toxicity). For the AUC-ROC curve, the

baseline denotes a random classifier, manifesting as a diagonal line with an AUC-ROC value of 0.5. Conversely, the PR curve's baseline reflects a situation where all classifications are assumed to be positive, resulting in a horizontal line on the precision-recall plot; the position of this line on the Y-axis is contingent upon the characteristics of the data under consideration [36,37]. These prediction metrics are calculated and reported for the training (without the SVM-SMOTE generated synthetic samples used for data balancing) and withheld test data (with the missing data-imputed for both). For the purposes of this work, the authors have considered the top ML models as the ones with the highest test data F1 score. Confidence intervals of 95% were calculated using bootstrap resampling with replacement over 1000 iterations. In each iteration, the model was retrained using balanced data, and metrics were recalculated on the unbalanced training and testing datasets. The standard deviation of the metric values across iterations was used to estimate the confidence interval under the assumption of a normal distribution [38].

## Sequential feature selection

In various domains, including healthcare, datasets may exhibit high dimensionality, referring to the presence of a large number of variables or features. This characteristic can adversely affect the development and interpretability of some machine learning algorithms (Logistic Regression, Support Vector Machines, K Nearest Neighbors, and Gaussian Naive Bayes) [39,40]. To reduce dimensionality, several approaches exist such as feature extraction and feature selection [41]. In this work, multiple variations of sequential feature selection were initially considered, including Sequential Forward Selection (SFS), Sequential Backward Selection (SBS), Sequential Forward Floating Selection (SFFS) and Sequential Backward Floating Selection (SBFS), which used as their estimator the same model to later be used for training [42–44]; after experimentation, Sequential Forward Selection was chosen and applied during the training of LR, SVM, KNN and GNB due to faster computation times and better performance metrics. Note that, regardless of traditional statistical analysis, both marginally significant variables, and those that were not, are explored when training the ML algorithms. The forward feature selection process adds one feature into the model at a time, determining inclusion based on which predictor optimizes the evaluation criteria, which in our case was the F1 score. As part of model training, a 10-fold Stratified Shuffle Split cross-validator was used over the class-balanced training data to reduce overfitting and appropriately assess the performance metrics of the sequential feature selection algorithm [45–46]. A SHAP analysis [47] or variable importance plot (For RF) was done for the top 3 models to study the relevance of the final features that were included.

## Machine learning algorithms

A total of 7 machine learning models were implemented and compared. The included models were the following: Logistic Regression (LR), Random Forest (RF), K-Nearest Neighbors, Support Vector Machines (SVM), Gaussian Naive Bayes (GNB), Multi-Layer Perceptron Neural Network, and XGBoost (XGB). While there are many other ML classification algorithms in the literature, these seven choices represent the most commonly utilized algorithms for radiation outcome modeling [48]. The selection of these methods provided a range from simpler models (such as LR, KNN, and GNB) to more complex approaches (such as RF, SVM, MLP and XGB), addressing practical challenges related to dataset size and potential risks of overfitting. The baseline for the precision – recall curve was determined to be a horizontal line equal to 0.237 based on a classifier that labels all predictive instances as positive within the held-out testing data. After selecting the most relevant features through the Sequential Feature Selection process for the appropriate models, the hyperparameters of all 7 models were further fine-tuned by using a Grid Search over another 10-fold Stratified Shuffle Split cross validator to optimize prediction under each model choice; the hyperparameter search space used by Grid Search is detailed in S1 Table. The Python code and study database have been made available to the reader. This data is available at: https://github.com/AndresPB95/ML-Model-Gynecological-HDR-G3Plus-Toxicities. A comprehensive diagram depicting the full machine learning workflow is provided in Fig 1, and S2 Table presents a summary of all the features explored by the

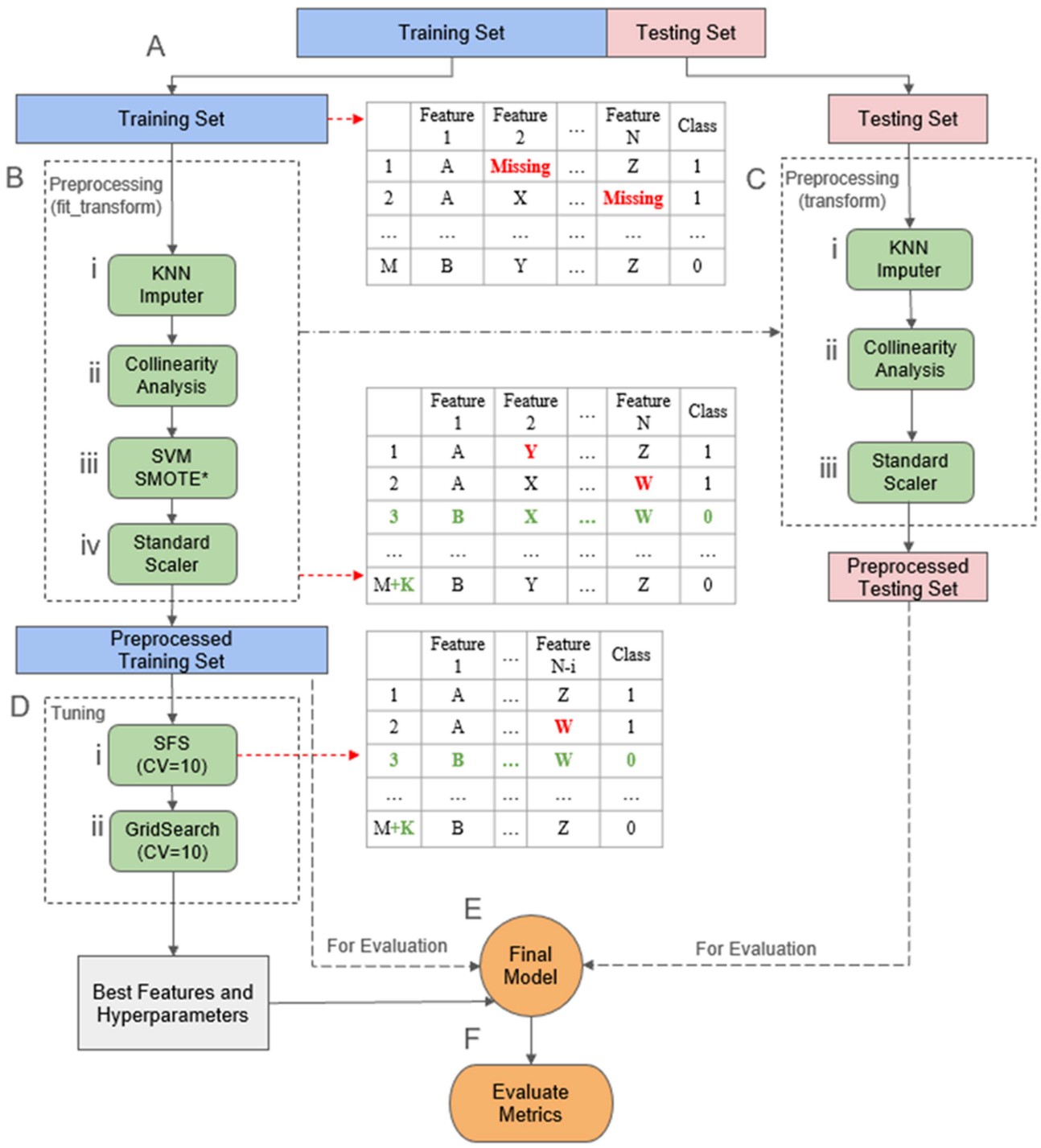

**Fig 1. Flowchart outlining the steps used when training and evaluating the different models.** The process is divided into the following steps: **(A) Initial Train/Test split**: The data is initially divided into training and testing sets. The training set is used for most of the model development process, while the testing set is reserved to simulate new, unseen data. **(B) Data preprocessing (Training Set)**: Preprocessing steps include: (i) A *KNN Imputer* is fitted and applied to the training data to fill in missing values, (ii) *Collinear* features are removed, (iii) *SVM SMOTE* is used to oversample the positive class (*Only for training). Note: A separate, unbalanced copy of the training set was retained for evaluation, (iv) a *StandardScaler* is fitted and applied to the training data ensuring they are on a comparable scale. **(C) Data preprocessing (Testing Set)**: The preprocessing objects fitted to the training set are subsequently applied to the testing set: (i) The *KNN Imputer* is used to fill in missing values in the testing data, (ii) *Collinear* features are removed,

(iii) the *StandardScaler* is applied for normalization. Note: SVM SMOTE was NOT used to oversample the test set. **(D) Hyperparameter Tuning**: For each model, the following tuning procedures are conducted using 10-fold cross-validation: (i) *Sequential Feature Selection* (if applicable) creates and trains multiple models by adding one feature at a time. Each model's F1 score is tested by comparing the predicted values with the known labels, and features that improve the F1 score are retained, building towards the most effective feature set. (ii) *GridSearch* trains multiple models with various hyperparameter combinations. Each combination's F1 score is tested by comparing the predicted values with the known labels, and the best-performing combination is selected for the final model. **(E) Final Model Generation**: After identifying the optimal hyperparameters and features, a final model is trained using the entire balanced training set. **(F) Evaluation**: The model's performance is evaluated by comparing its predictions against the known labels using both the unbalanced training set and the testing set.

ML models, along with their types. The initial features included in the database of study were selected through discussions with physicians and professionals directly involved in HDR treatments.

## Results

The dataset included demographic and clinical data for n = 233 patients, of which n = 56 (24%) had a grade 3 or higher toxicity. It comprised 32 features: 3 ordinal, 7 categorical, and 22 numerical variables. The highest missing data rates were observed for the total number of treatment days (11%) and maximum tumor length (16.7%), while all other features had missingness rates of 4.3% or less. The demographic, treatment, and tumor-related data are shown in Table 1. Patients who experienced grade 3 or higher toxicity were found to have longer follow-up (median 12.4 months versus 3.8 months), more likely to have low or very high comorbidity scores and had significantly higher HR-CTV values (median 50 cc versus 39 cc, p = 0.041).

Table 2 compares median dose coverage to the tumor (V100%, D50%, D90%, and D98%) and the doses to the organs at risk (OARs) by toxicity status. Patients with grade 3 or higher toxicities had significantly higher D2cc doses to the rectum (p = 0.043), but no other doses were statistically significantly different. The HR-CTV V100, D1cc - Rectum, and doses to the sigmoid colon were slightly higher for the group with grade 3 or higher toxicities but not statistically significant.

The seven machine learning models were then fitted using all variables included in Tables 1 and 2 as described in the Methods section. The performance of these models on the withheld test data are depicted visually in Figs 2 and 3. Numeric comparisons based on both the (class-imbalanced) training data and withheld test data are shown in Table 3. The top three models for predicting grade 3 or higher toxicities are found to be Support Vector Machines (SVM), Random Forests (RF), and Logistic Regression (LR) with F1 testing scores of 0.63, 0.57 and 0.52, normMCC testing scores of 0.75, 0.77 and 0.71, and Accuracy testing scores of 0.80, 0.85 and 0.81, respectively. All values shown in Table 3 assume a classification threshold value of 0.5 for toxicity prediction. Note that this table also includes the metrics from the training data, which for some models (MLP and KNN) disagree strongly with the test data performance measures, indicating severe overfitting in the training data. Table 4 exhibits the most relevant features and the values of the hyperparameters selected by the GridSearchCV optimization algorithm over the training data. The top features repeated among these three models are Chemotherapy, Charlson Comorbidity Index, KPS, D2cc - Small Bowel, Stage, Histology, and Follow-Up Time.

The total runtime for the training of Logistic Regression, Random Forest, K-Nearest Neighbors, Support Vector Machines, Gaussian Naive Bayes, Multi-Layer Perceptron, and XGBoost was 0.35, 8.24, 0.45, 0.73, 0.13, 104.32, and 28.86 minutes, respectively. The Python script was executed on a computer equipped with a 14th Generation Intel(R) Core(TM) i9-14900HX 2.20 GHz processor (E-cores up to 4.10 GHz P-cores up to 5.80 GHz), 32 GB of DDR5–5600 MHz (SODIMM) RAM, and a 64-bit Windows 11 Pro operating system.

## Discussion

This study aimed to investigate the utility of using machine learning models to predict grade 3 or higher toxicities in gynecologic cancer patients treated with EBRT and interstitial or T&O brachytherapy. The database was analyzed using traditional statistics which compared groups with and without grade 3 + toxicities; disease free survival and local control were

**Table 1. Comparison of patient, treatment, and tumor characteristics between groups with and without grade 3 or higher toxicities.**

| | | Full Cohort | | No Grade 3+ Toxicity | | Grade 3+ Toxicity | | |
|---|---|---|---|---|---|---|---|---|
| | | n = 233 | 100% | n = 177 | 76% | n = 56 | 24% | p-value |
| Length of Follow-Up (mo) | | 6.1 | IQR: [1.4–18.2] | 3.8 | IQR: [1.2–16.4] | 12.4 | IQR: [7.1–22.1] | **< 0.001** |
| Age at Completion | | 53.6 | STD: 14.8 | 54.4 | STD: 14.7 | 50.8 | STD: 14.8 | 0.107 |
| Non-Caucasian | | 25 | 11% | 17 | 10% | 8 | 14% | 0.328 |
| BMI | | 28.0 | STD: 8.3 | 28.0 | STD: 8.6 | 27.9 | STD: 7.6 | 0.969 |
| Charlson Comorbidity Index | | | | | | | | **0.014** |
| | Low [0–2] | 87 | 37% | 60 | 34% | 27 | 48% | |
| | Medium [3] | 43 | 18% | 33 | 19% | 10 | 18% | |
| | High [4–5] | 60 | 26% | 54 | 31% | 6 | 11% | |
| | Very High [>5] | 43 | 18% | 30 | 17% | 13 | 23% | |
| KPS | | | | | | | | 0.369 |
| | Good [90–100] | 147 | 63% | 115 | 65% | 32 | 57% | |
| | Normal [80] | 62 | 27% | 43 | 24% | 19 | 34% | |
| | Bad [50–70] | 23 | 10% | 18 | 10% | 5 | 9% | |
| Treatment Days | | 60 | IQR: [52–71] | 60 | IQR: [52–69] | 61 | IQR: [52–74] | 0.504 |
| Applicator: T&O | | 80 | 34% | 63 | 36% | 17 | 30% | 0.521 |
| Concurrent Chemo | | 201 | 86% | 153 | 86% | 48 | 86% | 1.000 |
| Type of Boost | | | | | | | | 0.681 |
| | None | 139 | 60% | 108 | 61% | 31 | 55% | |
| | Sequential | 54 | 23% | 39 | 22% | 15 | 27% | |
| | SIB | 40 | 17% | 30 | 17% | 10 | 18% | |
| Tumor Size (cm) | | 5.4 | STD: 2.1 | 5.4 | STD: 2.0 | 5.6 | STD: 2.5 | 0.622 |
| HR-CTV (cc) | | 43 | IQR: [27–74] | 39 | IQR: [25–71] | 50 | IQR: [34–77] | **0.041** |
| Tumor Site | | | | | | | | 0.864 |
| | Cervix | 194 | 83% | 147 | 83% | 47 | 84% | |
| | Endometrium | 16 | 7% | 13 | 7% | 3 | 5% | |
| | Other | 23 | 10% | 17 | 10% | 6 | 11% | |
| Cancer Stage | | | | | | | | 0.163 |
| | Stage 1 | 45 | 19% | 37 | 21% | 8 | 14% | |
| | Stage 2 | 57 | 25% | 41 | 23% | 16 | 29% | |
| | Stage 3 | 107 | 46% | 84 | 48% | 23 | 41% | |
| | Stage 4 | 22 | 10% | 13 | 7% | 9 | 16% | |
| Histology: SCC | | 180 | 77% | 136 | 77% | 44 | 79% | 0.857 |
| MRI Fused | | 105 | 45% | 84 | 47% | 21 | 38% | 0.219 |

also reported (S1 Fig). To design the toxicity prediction models, data were encoded and pre-processed. Next, a sequential feature selector method was used when appropriate, and hyperparameter tuning was performed.

From a clinical application standpoint, ML models can significantly enhance patient care by identifying individuals who are at higher risk of developing severe toxicities. Clinicians can use this predictive information to modify the treatment plan to reduce the risk of severe toxicities, tailor patient counseling, set appropriate expectations, and implement proactive monitoring strategies. For example, patients identified as high-risk may benefit from more frequent follow-up appointments, early interventions for symptom management, or a change in the prescription dose. Additionally, the model can aid in shared decision-making by involving patients in discussions about the potential risks and benefits of their treatment plans. By stratifying patients based on their predicted toxicity risk, healthcare teams can allocate resources more

**Table 2. HR-CTV and OAR dosimetric values between groups with and without grade 3 or higher toxicities.**

| | Full Cohort | | No Grade 3+ Toxicity | | Grade 3+ Toxicity | | |
|---|---|---|---|---|---|---|---|
| | N = 233 | (100%) | N = 177 | (76%) | N = 56 | (23%) | |
| | Median | IQR | Median | IQR | Median | IQR | p-value |
| HR-CTV V100 (cc) | 80.1 | [54.3 - 129.3] | 74.5 | [50.8 - 123.3] | 88.3 | [60.3 - 132.7] | 0.104 |
| HR-CTV D50 (Gy) | 110.0 | [101.2 - 119.1] | 110.8 | [101.4 - 119.1] | 109.8 | [100.3 - 118.9] | 0.628 |
| HR-CTV D90 (Gy) | 83.1 | [79.9 - 87.7] | 83.1 | [80.0 - 87.7] | 83.1 | [79.2 - 86.7] | 0.967 |
| HR-CTV D98 (Gy) | 75.3 | [70.9 - 79.9] | 75.1 | [70.9 - 79.8] | 75.7 | [69.7 – 80.0] | 0.837 |
| D0.1cc - Bladder (Gy) | 97.5 | [83.4 - 114.5] | 97.0 | [83.3 - 112.9] | 98.6 | [85.2 - 120.8] | 0.455 |
| D1cc - Bladder (Gy) | 84.9 | [74.9 - 95.4] | 84.6 | [75.2 - 94.7] | 85.5 | [74.6 - 98.1] | 0.570 |
| D2cc - Bladder (Gy) | 79.7 | [71.3 - 89.2] | 79.2 | [71.8 - 88.5] | 81.9 | [70.5 - 91.1] | 0.569 |
| D0.1cc - Small Bowel (Gy) | 67.2 | [52.8 - 83.8] | 68.0 | [52.7 - 84.5] | 65.2 | [53.9 - 78.2] | 0.838 |
| D1cc - Small Bowel (Gy) | 60.9 | [51.5 - 73.5] | 61.1 | [51.2 - 73.7] | 59.6 | [52.4 - 70.1] | 0.852 |
| D2cc - Small Bowel (Gy) | 59.3 | [50.7 - 69.6] | 59.4 | [50.6 – 70.0] | 57.8 | [51.0 - 66.4] | 0.898 |
| D0.1cc - Sigmoid Colon (Gy) | 74.3 | [62.0 – 86.0] | 73.9 | [59.9 - 86.3] | 76.6 | [65.2 - 85.4] | 0.277 |
| D1cc - Sigmoid Colon (Gy) | 67.0 | [57.2 - 75.4] | 66.5 | [56.1 – 75.0] | 69.1 | [60.3 - 76.3] | 0.182 |
| D2cc - Sigmoid Colon (Gy) | 64.2 | [55.3 - 71.6] | 63.8 | [54.3 - 71.2] | 65.6 | [58.0 - 72.2] | 0.172 |
| D0.1cc - Rectum (Gy) | 80.8 | [74.3 - 87.6] | 80.7 | [74.0 - 87.6] | 82.3 | [76.8 - 86.9] | 0.361 |
| D1cc - Rectum (Gy) | 71.4 | [66.2 - 78.4] | 71.1 | [65.7 - 76.6] | 73.7 | [68.5 - 79.9] | 0.066 |
| D2cc - Rectum (Gy) | 67.6 | [62.5 - 74.4] | 67.3 | [62.2 - 72.6] | 71.1 | [65.3 - 75.9] | *0.043* |

efficiently and personalize supportive care measures to mitigate adverse effects and maintain or improve patients' QoL. Thus, ML models can serve as a valuable tool for enhancing patient outcomes through individualized clinical management. Once validated, deployment of the best model into clinical workflows could be done with a decision-support application that provides real-time risk assessments during treatment planning. This tool could be implemented directly within the Eclipse environment via ESAPI scripting (C#), as a standalone browser-based application accessible to clinicians, or through alternative deployment methods adapted to the specific needs and infrastructure of each clinic.

A comparison of the patients with and without grade 3 toxicities, using basic marginal statistical analysis, suggested few differences between the groups including HR-CTV, Charlson Comorbidity Index, Length of Follow Up, and D2cc - Rectum. Some of these variables such as the HR-CTV and D2cc - Rectum have been previously shown to be predictors of grade 3 or higher toxicities of HDR brachytherapy. Lee et al. observed that patients with grade 3–4 toxicities had a significantly higher median HR-CTV of 111 cc compared to 43 cc for those patients with grade 0–2 toxicities [49]. Mesko et al. found a statistically significant difference between patients with and without grade 3 toxicities, with median HR-CTV values of 93.8 cc and 51 cc, respectively [50]. Mazeron et al. found that rectal D2cc values equal to or greater than 75 Gy EQD2 are associated with higher grade and more frequent toxicities in MRI-guided adaptive brachytherapy for locally advanced cervical cancer [51]. When compared with traditional statistics, machine learning models consider nonlinear interactions between variables [52]. Our top-performing model selected 10 features, with Length of Follow-Up, D2cc to the Small Bowel, Charlson Comorbidity Index, HR-CTV volume, and D2cc to the sigmoid being the most relevant (S2 Fig), agreeing with the discussed literature and the statistical analysis. Notably, Length of Follow-Up and Charlson Comorbidity Index were also significant predictors in our second and third best-performing models (S3 and S4 Figs). Higher D2cc doses to the small bowel and sigmoid indicate greater radiation exposure, increasing the risk of severe toxicity, while a higher Charlson Comorbidity Index reflects poorer overall health and greater susceptibility to adverse effects. Larger HR-CTVs may necessitate more aggressive treatment, heightening toxicity risk, and Length of Follow-Up is crucial for capturing late-onset toxicities. It is important to note that the features chosen by SFS may exclude variables that are easily manipulable when creating a treatment plan, particularly dosimetric variables. This issue could be explained twofold: 1) certain

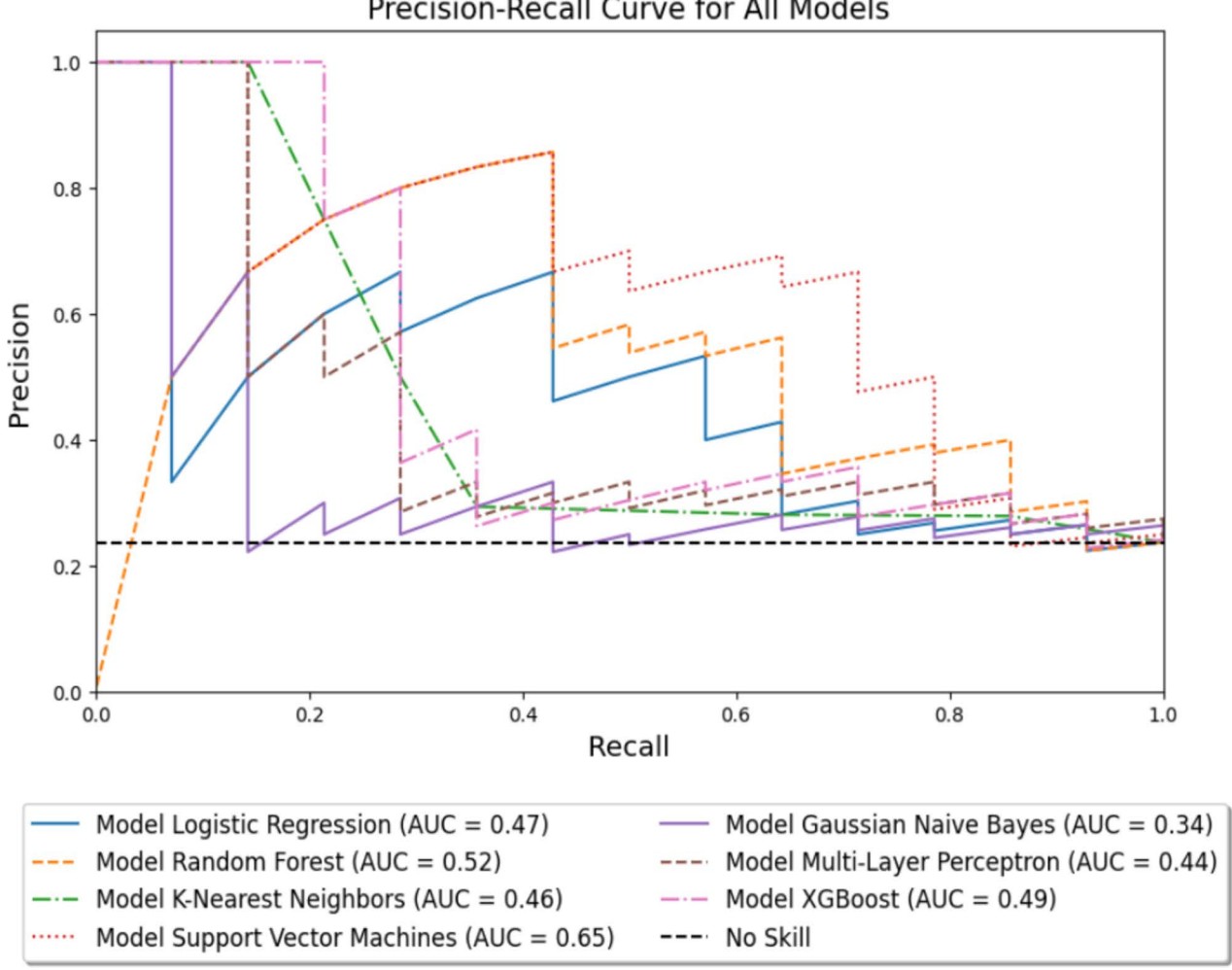

**Fig 2. Precision-Recall curves comparing 7 machine learning models and a baseline value.** PR curves are computed using the withheld test data. SVM is the model with the highest area under the curve.

combinations of hyperparameters could limit the ability of SFS to find the correct interactions between features in the final selection; or 2) certain combinations of features could be more relevant and produce better predictions than when using actual dosimetric data. A model without any dosimetric features would still be useful for predicting toxicity risk but would not provide the clinician the option of adjusting the treatment plan to reduce the risk of toxicity.

Supervised machine learning has been utilized to perform classification tasks in various areas of healthcare such as for predicting diagnosis and prognosis of COVID-19 patients, prediction of hospitalization due to heart disease, and outcome prediction of infectious diseases [53–55]. To the authors' knowledge, this is the first analysis using and comparing multiple models for predicting grade 3 or higher toxicities in gynecologic cancer patients treated with external beam radiation and HDR interstitial or T&O brachytherapy. Through March 2020, there were only 53 published studies on the use of machine learning to predict radiation-induced toxicities [56], and through September 2023, only 14 studies had been published on deep learning models to predict toxicities from radiation treatment [57].

Regarding ML in brachytherapy toxicity prediction, Tian et al. developed a model for predicting fistula formation, reporting a recall of 97.1% and AUC of 0.904 utilizing the SMOTE algorithm and a SVM model with a radial basis kernel function on

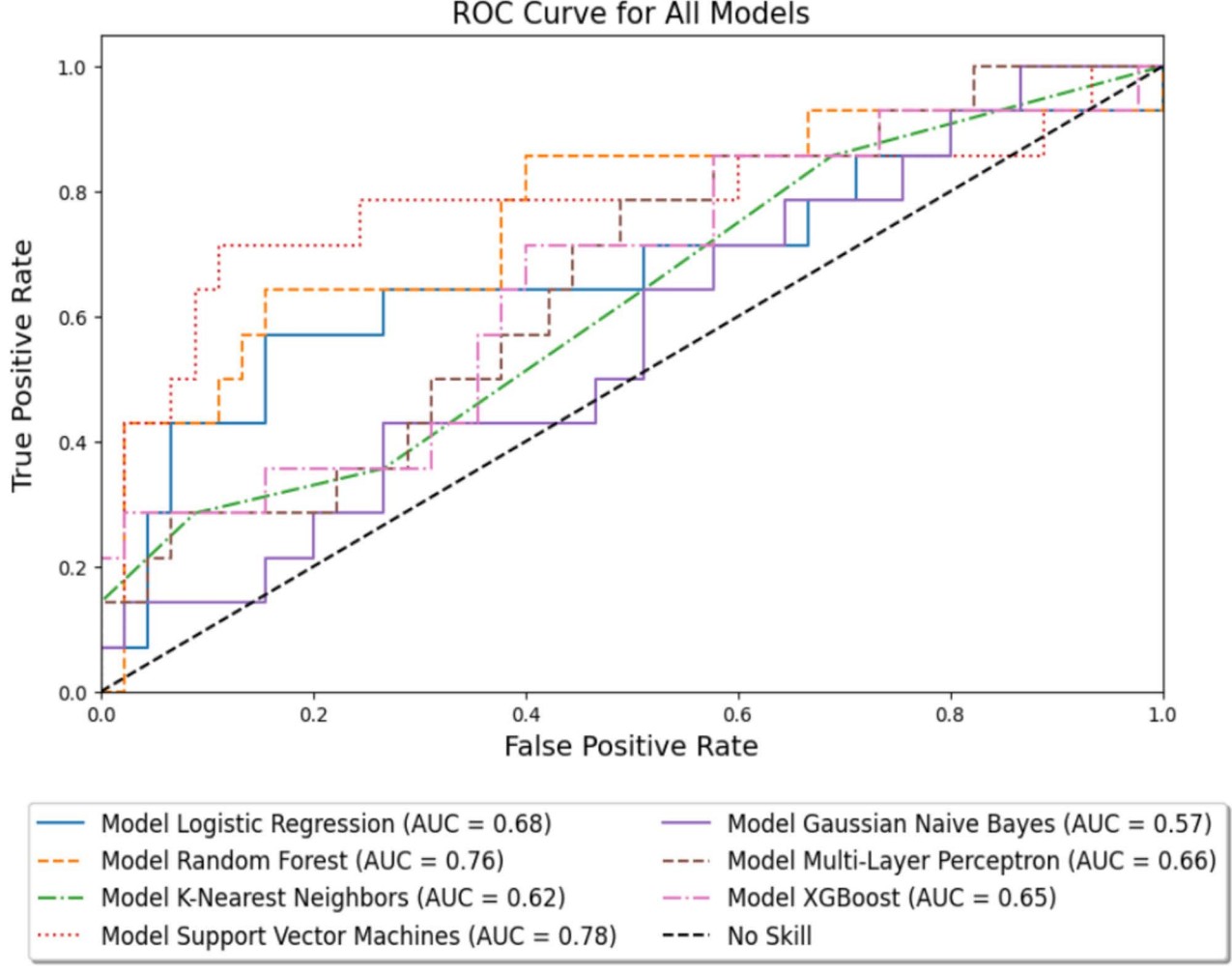

**Fig 3. Receiver Operating Characteristics curves for 7 machine learning models and a baseline value.** ROC curves are computed using the withheld test data. SVM is the model with the highest area under the curve.

a database that included 35 patients with 7 positive cases; the limitation of this study lies in the small dataset, no withheld test dataset, high risk of model overfitting, and only using one model in their study [58]. For prediction of rectal toxicities, Chen et al. and Zhen et al. predicted grade 2 or higher rectal toxicity by using SVM and convolutional neural networks, respectively, with scores of (cross-validation estimated) recall and AUC of 0.85 and 0.91 for the former and 0.75 and 0.89 for the latter. Their work includes the addition of dose map features for the training of the model; both of these studies were done with a database of 42 patients with 12 positive cases of patients that developed toxicities [59,60]. Additionally, there has been work by Lucia et al. who developed Normal Tissue Complication Probability (NTCP) models for acute and late gastrointestinal, genitourinary, and vaginal toxicities using a database of 102 patients that included radiomic features, but only for a logistic regression model, which obtained balanced accuracy scores between 63.99 and 78.41 [61]. Cheon et al. considered deep learning models for predicting late bladder toxicities which outperformed its multivariable logistic regression counterpart [62], with data of 281 patients which achieved an F1 score of 0.76. In contrast to earlier investigations, our study presents the largest patient dataset used for predicting grade 3 or higher toxicities. Similar to these studies, we employ data-balancing algorithms to promote stability in the model training stage. Our methodology incorporates feature

**Table 3. Training and testing performance metrics for the considered machine learning models.**

| Model | Metric | Training Dataset | | Testing Dataset | |
|---|---|---|---|---|---|
| | | Mean | 95% CI | Mean | 95% CI |
| SVM | F1 | 0.61 | [0.55 - 0.67] | **0.63** | [0.51 - 0.74] |
| | Accuracy | 0.82 | [0.74 - 0.89] | 0.8 | [0.69 - 0.91] |
| | normMCC | 0.75 | [0.70 - 0.79] | 0.75 | [0.65 - 0.84] |
| | Precision | 0.63 | [0.55 - 0.70] | 0.56 | [0.46 - 0.66] |
| | Recall | 0.6 | [0.46 - 0.73] | **0.71** | [0.54 - 0.89] |
| | AUC-ROC | 0.87 | [0.83 - 0.91] | **0.78** | [0.68 - 0.87] |
| | AUC-PR | 0.6 | [0.51 - 0.68] | **0.65** | [0.50 - 0.80] |
| RF | F1 | 0.82 | [0.75 - 0.89] | 0.57 | [0.39 - 0.75] |
| | Accuracy | 0.92 | [0.86 - 0.97] | **0.85** | [0.74 - 0.95] |
| | normMCC | 0.89 | [0.84 - 0.94] | **0.77** | [0.65 - 0.89] |
| | Precision | 0.89 | [0.79 - 0.99] | **0.86** | [0.70 - 1.00] |
| | Recall | 0.76 | [0.67 - 0.86] | 0.43 | [0.18 - 0.68] |
| | AUC-ROC | 0.97 | [0.94 - 1.00] | 0.76 | [0.65 - 0.87] |
| | AUC-PR | 0.93 | [0.86 - 1.00] | 0.52 | [0.35 - 0.69] |
| LR | F1 | 0.45 | [0.40 - 0.50] | 0.52 | [0.43 - 0.61] |
| | Accuracy | 0.75 | [0.68 - 0.81] | 0.81 | [0.73 - 0.89] |
| | normMCC | 0.64 | [0.60 - 0.69] | 0.71 | [0.65 - 0.78] |
| | Precision | 0.47 | [0.42 - 0.53] | 0.67 | [0.56 - 0.77] |
| | Recall | 0.43 | [0.33 - 0.53] | 0.43 | [0.31 - 0.55] |
| | AUC-ROC | 0.72 | [0.68 - 0.76] | 0.68 | [0.63 - 0.72] |
| | AUC-PR | 0.43 | [0.38 - 0.48] | 0.47 | [0.38 - 0.56] |
| XGB | F1 | 1.00 | [0.94 - 1.00] | 0.48 | [0.34 - 0.62] |
| | Accuracy | 1.00 | [0.96 - 1.00] | 0.78 | [0.70 - 0.86] |
| | normMCC | 1.00 | [0.96 - 1.00] | 0.67 | [0.58 - 0.77] |
| | Precision | 1.00 | [0.91 - 1.00] | 0.55 | [0.42 - 0.67] |
| | Recall | 1.00 | [0.93 - 1.00] | 0.43 | [0.22 - 0.63] |
| | AUC-ROC | 1.00 | [0.97 - 1.00] | 0.71 | [0.64 - 0.79] |
| | AUC-PR | 1.00 | [0.90 - 1.00] | 0.49 | [0.34 - 0.64] |
| MLP | F1 | 1.00 | [0.93 - 1.00] | 0.39 | [0.26 - 0.52] |
| | Accuracy | 1.00 | [0.95 - 1.00] | 0.63 | [0.53 - 0.73] |
| | normMCC | 1.00 | [0.96 - 1.00] | 0.57 | [0.48 - 0.67] |
| | Precision | 1.00 | [0.91 - 1.00] | 0.32 | [0.21 - 0.43] |
| | Recall | 1.00 | [0.93 - 1.00] | 0.5 | [0.31 - 0.69] |
| | AUC-ROC | 1.00 | [0.95 - 1.00] | 0.66 | [0.54 - 0.77] |
| | AUC-PR | 1.00 | [0.85 - 1.00] | 0.44 | [0.31 - 0.57] |
| KNN | F1 | 0.72 | [0.66 - 0.78] | 0.32 | [0.19 - 0.46] |
| | Accuracy | 0.86 | [0.81 - 0.91] | 0.64 | [0.53 - 0.76] |
| | normMCC | 0.81 | [0.77 - 0.86] | 0.54 | [0.44 - 0.65] |
| | Precision | 0.68 | [0.62 - 0.74] | 0.29 | [0.19 - 0.40] |
| | Recall | 0.76 | [0.66 - 0.87] | 0.36 | [0.15 - 0.57] |
| | AUC-ROC | 0.91 | [0.87 - 0.95] | 0.62 | [0.51 - 0.73] |
| | AUC-PR | 0.78 | [0.71 - 0.85] | 0.46 | [0.34 - 0.57] |

*(Continued)*

**Table 3.** (Continued)

| Model | Metric | Training Dataset | | Testing Dataset | |
|-------|--------|------|---------|------|---------|
| | | Mean | 95% CI | Mean | 95% CI |
| GNB | F1 | 0.33 | [0.26 - 0.41] | 0.24 | [0.08 - 0.39] |
| | Accuracy | 0.79 | [0.73 - 0.86] | 0.78 | [0.67 - 0.89] |
| | normMCC | 0.66 | [0.61 - 0.71] | 0.62 | [0.51 - 0.72] |
| | Precision | 0.75 | [0.69 - 0.81] | 0.67 | [0.55 - 0.79] |
| | Recall | 0.21 | [0.04 - 0.39] | 0.14 | [0.00 - 0.41] |
| | AUC-ROC | 0.67 | [0.62 - 0.72] | 0.57 | [0.47 - 0.67] |
| | AUC-PR | 0.4 | [0.33 - 0.47] | 0.34 | [0.23 - 0.45] |

**Table 4. Most important features as selected by the Sequential Feature Selection algorithm (where appropriate) and found optimal hyperparameters for the top 3 scoring models.**

| SVM | | RF | | LR | |
|-----|--|----|--|----|--|
| Features | Hyperparameters | Features | Hyperparameters | Features | Hyperparameters |
| Chemotherapy | C: 1 | All | n_estimators: 15 | Chemotherapy | C: 1 |
| Charlson | kernel: rbf | | max_features: log2 | Charlson | penalty: l2 |
| KPS | gamma: scale | | min_samples_leaf: 5 | KPS | solver: lbfgs |
| MRI | | | min_samples_split: 5 | Ethnicity | |
| D2cc Small Bowel | | | | Type of Boost | |
| D2cc Sigmoid | | | | Applicator | |
| Stage | | | | D2cc Small Bowel | |
| Histology | | | | D2cc Rectum | |
| HR-CTV | | | | Tumor Site | |
| Follow-Up Time | | | | Stage | |
| | | | | Histology | |
| | | | | Follow-Up Time | |

selection for all models except for MLP and RF. Specifically, we leverage the Sequential Feature Selection Algorithm to promote parsimony within the model fit. This aligns with the methodologies employed in previous reports.

Overfitting occurs when a model becomes overly complex, capturing noise in the training data instead of learning the underlying patterns, leading to poor predictions when applied to new data [63]. To mitigate this phenomenon, the use of a withheld testing data set is required to assess the degree of overfitting and the performance of the model [64]. The authors recommend that the training and withheld data testing scores should always be reported to provide a comprehensive understanding of a model's performance. A clear illustration of overfitting can be appreciated in Table 3 for the MLP, KNN and XGB models, where they achieved impressive training F1 scores of 1.00, 0.72 and 1.00, respectively; contrasting sharply with their testing scores of 0.39, 0.32, and 0.48. These scores show that these 3 models are not generalizable for predicting new similar data points. A likely explanation for these three models' performance would be their hyperparameter sensitivity. In the case of MLP, there are a large number of hyperparameters and possibilities for neuron activation; similarly, XGBoost offers many adjustable settings, and KNN relies on a highly accurate selection of k-neighbors to make a reliable classification [65]. Each model merits a robust and thorough search space to cover sufficient model possibilities and an analysis of different hyperparameter tuning packages.

Regarding the scoring metrics, our study showed that the support vector machine was the best model for predicting grade 3 toxicities, obtaining a training F1 score of 0.61, accuracy of 0.82, normMCC of 0.75, precision of 0.63, recall of

0.6, AUC-ROC of 0.87, and AUC-PR of 0.6; whereas for that same model, the test data obtained an F1 score of 0.63, accuracy of 0.80, normMCC of 0.75, precision of 0.56, recall of 0.71, AUC-ROC of 0.78, and AUC-PR of 0.65. In the withheld test data, out of all the patients that had a toxicity (n = 14), 71% were correctly predicted by the model (TP = 10); and out of all the predicted cases, 56% represented a true toxicity event and were not false positives (FP = 8). Given the high level of clinical uncertainty in whether patients will develop toxicities, this may be viewed as an adequate performance; while slightly lower, it remains comparable to similar studies, likely reflecting the increased difficulty associated with accurately predicting higher-grade (Grade ≥3) toxicities. An important detail that must be considered is that the precision value is as important as the recall, since during normal clinical practice it is equally as important to avoid false positives as it is to detect true positive cases. In particular, a toxicity prediction model may suggest that the physician consider lowering the dose to certain OARs to prevent these high-grade radiation-related side effects; an algorithm with good recall but prone to predicting false positives may lead to reducing the dose for a patient not susceptible to developing toxicities. This reduction, in turn, may involve sacrificing a portion of tumor coverage, potentially decreasing tumor control. For this reason, the F1 score emerges as the optimal metric for evaluating the model's performance. In future investigations within this area, prioritizing either the recall or the precision score, which is not replaceable by specificity, could be explored. Notably, specificity becomes less valuable in situations marked by an imbalance with a majority of true negatives [66] as the model's ability to predict negative outcomes can render overly optimistic scores in such scenarios. Once a best performing model has been identified, multi-institutional clinical trials will be needed to assess their performance on routine clinical practice.

The strength of this work lies in several key aspects. First, the study analyzes multiple machine learning models to find the best fit across a variety of common prediction algorithms. Additionally, we divide the data into training and testing sets before employing cross-validation for the model's training, enhancing generalizability of the final models and providing more trustworthy measures of out-of-sample performance, despite potential reductions in the values of these metrics. The use of a Stratified Shuffle Split approach guarantees that there will be a positive class on the testing set of the cross validation, ensuring meaningful performance in every split. Furthermore, the focus on the F1 score and reporting precision as the performance metrics is of practical relevance for assessing the clinical performance of the model, especially when predicting toxicities.

The primary limitation of this study is the relatively small sample size. Although our dataset of 233 data points is larger than those used in previous studies, only 75% of this data was utilized for training, which may influence the model's accuracy and introduce uncertainty. Research has shown that smaller datasets are more prone to overfitting and can be influenced by random variability in the data [67]. To address this, several models were evaluated, ranging from simpler methods to more complex machine learning algorithms. The authors plan on elaborating multi-institutional studies to address sample size limitations in the future. Additionally, only the dosimetric, treatment, and tumor variables were considered in this study, but not any additional features such as dose maps with spatial information. Regarding data balancing through Synthetic Oversampling, alternative techniques like threshold tuning could be investigated.

Furthermore, developing methods to address overfitting could be beneficial, particularly for MLP, KNN, and XGBoost. For KNN, different alternatives for the classifier could be explored, such as weighted-KNN, Radius-based Nearest Neighbor, or changing to a different distance metric other than Euclidean. For MLP, a much more defined hyperparameter space will be addressed in future studies. Several overfitting solutions have been outlined by Ying [68], such as Early-Stopping, L1 and L2 regularization parameters, etc. Whereas for XGBoost, the number of trees and its depth could be further limited, as well as manipulating some other relevant hyperparameters.

Finally, more computationally intensive approaches could be explored in future research. These include performing comprehensive hyperparameter optimization using dedicated libraries such as Optuna [69], evaluating additional ensemble-based methods like AdaBoost, or combining multiple algorithms. Other promising directions include the implementation of Multi-Feature Combined models as described by Yang et al. [70], as well as physics-informed ML [71]

approaches that incorporate fundamental brachytherapy dose calculations to provide a physics-based foundation for data-driven model refinements.

## Conclusion

Multiple machine learning models were trained and assessed to predict grade 3 or higher toxicity development in patients with gynecologic malignancies who received EBRT and interstitial or T&O brachytherapy treatment yielding satisfactory results for the top performing model. This novel approach of toxicity prediction holds the potential to set a new paradigm in standard clinical care and contribute towards personalized care in radiation therapy. New techniques to improve model training need to be explored, and overcoming machine learning limitations like small datasets requires collaborative efforts among peers. In the future, further investigations are needed to prospectively validate these models in other healthcare settings.

## Supporting information

**S1 Fig. Kaplan Meier plots for the entire patient cohort.** A) Disease Free Survival and B) Local control.
(PDF)

**S2 Fig. SHAP analysis for Support Vector Machine.**
(PDF)

**S3 Fig. Variable Importance plot for Random Forest.**
(PDF)

**S4 Fig. SHAP analysis for Logistic Regression.**
(PDF)

**S1 Table. Hyperparameter Search Space and MLP architecture.**
(PDF)

**S2 Table. Summary of input features and output from models.** The variable type and number of missing data points for each input is shown.
(PDF)

## Author contributions

**Conceptualization:** Andres Portocarrero-Bonifaz, Keith T. Sowards, Scott R. Silva.

**Data curation:** Andres Portocarrero-Bonifaz, Salman Syed, Maxwell Kassel, Grant W. McKenzie, Vishwa M. Shah, Bryce M. Forry, Jeremy T. Gaskins, Thulasi Babitha Avula, Adrianna Masters, Jose G. Schneider, Scott R. Silva.

**Formal analysis:** Andres Portocarrero-Bonifaz, Jeremy T. Gaskins, Jose G. Schneider.

**Investigation:** Andres Portocarrero-Bonifaz, Salman Syed, Vishwa M. Shah, Bryce M. Forry, Jeremy T. Gaskins, Keith T. Sowards, Thulasi Babitha Avula, Adrianna Masters, Jose G. Schneider, Scott R. Silva.

**Methodology:** Andres Portocarrero-Bonifaz, Salman Syed, Maxwell Kassel, Grant W. McKenzie, Vishwa M. Shah, Bryce M. Forry, Jeremy T. Gaskins, Keith T. Sowards, Thulasi Babitha Avula, Adrianna Masters, Jose G. Schneider, Scott R. Silva.

**Project administration:** Andres Portocarrero-Bonifaz, Scott R. Silva.

**Resources:** Andres Portocarrero-Bonifaz.

**Software:** Andres Portocarrero-Bonifaz, Maxwell Kassel, Jose G. Schneider.

**Supervision:** Andres Portocarrero-Bonifaz, Grant W. McKenzie, Jeremy T. Gaskins, Keith T. Sowards, Adrianna Masters, Scott R. Silva.

**Validation:** Andres Portocarrero-Bonifaz, Salman Syed, Maxwell Kassel, Grant W. McKenzie, Vishwa M. Shah, Bryce M. Forry, Jeremy T. Gaskins, Keith T. Sowards, Thulasi Babitha Avula, Adrianna Masters, Jose G. Schneider, Scott R. Silva.

**Visualization:** Andres Portocarrero-Bonifaz.

**Writing – original draft:** Andres Portocarrero-Bonifaz.

**Writing – review & editing:** Andres Portocarrero-Bonifaz, Salman Syed, Maxwell Kassel, Grant W. McKenzie, Vishwa M. Shah, Bryce M. Forry, Jeremy T. Gaskins, Keith T. Sowards, Thulasi Babitha Avula, Adrianna Masters, Jose G. Schneider, Scott R. Silva.

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
