## [Decision Letter · Decision Letter 0]

19 Nov 2024

PONE-D-24-43669Advancing patient care: Machine learning models for predicting grade 3+ toxicities in gynecologic cancer patients treated with HDR brachytherapyPLOS ONE

Dear Dr. Portocarrero Bonifaz,

Thank you for submitting your manuscript to PLOS ONE. After careful consideration, we feel that it has merit but does not fully meet PLOS ONE’s publication criteria as it currently stands. Therefore, we invite you to submit a revised version of the manuscript that addresses the points raised during the review process.

We look forward to receiving your revised manuscript.

Kind regards,

Li Yang, M.D.

Academic Editor

PLOS ONE

Journal Requirements: When submitting your revision, we need you to address these additional requirements. 1. Please ensure that your manuscript meets PLOS ONE's style requirements, including those for file naming. The PLOS ONE style templates can be found at https://journals.plos.org/plosone/s/file?id=wjVg/PLOSOne_formatting_sample_main_body.pdf and https://journals.plos.org/plosone/s/file?id=ba62/PLOSOne_formatting_sample_title_authors_affiliations.pdf 2. In the online submission form, you indicated that "Data cannot be shared publicly because of privacy reasons. Data are available upon request to the Brachytherapy Director: Dr. Scott Silva (contact via e-mail at scott.silva@louisville.edu) for research who meet the criteria for access to confidential data." All PLOS journals now require all data underlying the findings described in their manuscript to be freely available to other researchers, either 1. In a public repository, 2. Within the manuscript itself, or 3. Uploaded as supplementary information.This policy applies to all data except where public deposition would breach compliance with the protocol approved by your research ethics board. If your data cannot be made publicly available for ethical or legal reasons (e.g., public availability would compromise patient privacy), please explain your reasons on resubmission and your exemption request will be escalated for approval.

Reviewers' comments:

Reviewer's Responses to Questions

**Comments to the Author**

1. Is the manuscript technically sound, and do the data support the conclusions?

Reviewer #1: Yes

Reviewer #2: Yes

Reviewer #3: Yes

2. Has the statistical analysis been performed appropriately and rigorously? 

Reviewer #1: Yes

Reviewer #2: Yes

Reviewer #3: Yes

3. Have the authors made all data underlying the findings in their manuscript fully available?

Reviewer #1: Yes

Reviewer #2: Yes

Reviewer #3: Yes

4. Is the manuscript presented in an intelligible fashion and written in standard English?

Reviewer #1: Yes

Reviewer #2: Yes

Reviewer #3: Yes

5. Review Comments to the Author

Reviewer #1: This manuscript utilizes various machine learning methods to identify clinical features predicting toxicities from HDR brachytherapy in gynecological cancers. The manuscript demonstrates a high level of completeness in its application of machine learning, making it a valuable reference.

Detailed comments are as follows:

Introduction: The introduction provides a thorough overview of the treatment background for gynecological cancers and discusses machine learning in detail. However, it lacks:

1. Background information on toxicity from brachytherapy in gynecological cancers (definitions, incidence, causes, outcomes, etc.).

2. An overview of current research advancements in predicting toxicity from brachytherapy.

Methods and Results: The methods section adequately describes data collection, machine learning method selection, presenting patient baseline information well. The use of Precision-Recall curves and ROC curves for model evaluation is commendable. However, the following points require attention:

1. Although the authors included a substantial number of cases from a single center over a long period, using only 233 samples for training (with 75% allocated for training) is relatively small. The limitations of sample size should be addressed in the discussion.

2. An explanation of the missing data situation is necessary, along with the rationale for choosing KNN among various imputation methods (advantages and disadvantages).

3. Although the manuscript selects features such as Chemotherapy, Charlson Comorbidity Index, KPS, D2cc - Small Bowel, Stage, Histology, and Follow-Up Time, it does not clarify how these indicators are balanced and utilized in predicting toxicity.

4. Could constructing a multi-feature combined model provide more precise predictions? If further analysis is not feasible, please acknowledge the limitations in the discussion.

Discussion: While the manuscript offers an extensive discussion on the technical aspects of machine learning, the clinical application section is notably lacking.

Reviewer #2: ---In Figure 1 the text is not readable.

---The purpose section should be emphasized better.

---Which software did you analyze with?

---Can you give information about analysis times? Also provide information about the computer on which you performed the analysis.

---Write the limitations of the methods.

---Add the limitations of the study and your suggestions to the results.

Reviewer #3: Summary

In this work, the authors used data from a retrospective study of 233 patients to train statistical and machine learning models to predict grade 3 or higher toxicities in gynecological cancer patients treated with brachytherapy. Overall, I thought the paper was well written, easily readable, and would be interesting to applied researchers in the fields of oncology and gynecology. My experience only allows me to comment on the validity of this work with respect to the data modeling strategies utilized. In my opinion, the methods are well chosen but could use further clarification. Specified points for potential improvement are below:

Minor Points

• Line 77: A useful citation for this line might be Breiman’s 2001 paper on “Statistical Modeling: The Two Cultures”.

• Beyond the sample size, either in the introduction or results section, it would be beneficial to report the total number of features, the number of features that are categorical, ordinal, or continuous, and if any features had a high level of missingness.

• Line 158: Please explain further why Target Encoding was chosen as opposed to using something like one-hot encoding.

• In Fig 1, it is shown that the Standard Scaler operation is applied to the training and testing data. Is the same standard scaling that was applied to the training set the same as the test set? If so, it could be helpful to clarify this. If not, this is a potential issue since the models were fitted on a different scale.

• Line 209 – Are the models listed in parentheses the only ones that SFS was applied to? Might want to clarify this more.

• Line 300 – Are there any conjectures that could be included as to why these models fit but the others don’t?

• Table 3 – Maybe bold values to show best scores.

• Line 389 – In the conclusion, regularization methods for MLP and KNN models could be mentioned to fix the issue of overfitting.

• S1 Table – RF predictors become more stable as the number of estimators increases. Why was 50 chosen as the largest number of trees as opposed to something on the order of 100s?

Major Points

• Some features have a fair amount of missingness (17% for maximum tumor length and 11% for Treatment Days). It would be worthwhile to further clarify why the specific imputation method was chosen. Usually, the method chosen to deal with missing data is based on the expected type of missingness (e.g. missing completely at random vs missing at random). Either way, the variables with large degrees of missingness don’t appear to be included in the top 3 final models so it doesn’t seem like the imputation method should have a large impact.

• The random forest model has all the features included. An interesting result to report could be a VarImp plot or some other variable importance metric to see if the important features align with the other models or the univariate analyses.

• To further strengthen the argument that the SVM, RF, and LR tend to perform best, it would be interesting to see a subset of the analysis repeated on resampled train/test splits of the original data. However, it is understandable if this would be too computationally expensive to include.

6. PLOS authors have the option to publish the peer review history of their article (what does this mean? ). If published, this will include your full peer review and any attached files.

**Do you want your identity to be public for this peer review?** For information about this choice, including consent withdrawal, please see our Privacy Policy .

Reviewer #1: No

Reviewer #2: No

Reviewer #3: No

---

## [Author Response · Author response to Decision Letter 0]

9 Dec 2024

Thank you for your valuable suggestions and remarks from the editor and reviewers. After incorporating your feedback our manuscript has greatly improved. We hope that the present version of the manuscript will be acceptable for publication, and we look forward to your feedback.

---

## [Decision Letter · Decision Letter 1]

19 Mar 2025

PONE-D-24-43669R1Advancing patient care: Machine learning models for predicting grade 3+ toxicities in gynecologic cancer patients treated with HDR brachytherapyPLOS ONE

Dear Dr. Portocarrero Bonifaz,

Thank you for submitting your manuscript to PLOS ONE. After careful consideration, we feel that it has merit but does not fully meet PLOS ONE’s publication criteria as it currently stands. Therefore, we invite you to submit a revised version of the manuscript that addresses the points raised during the review process.

We look forward to receiving your revised manuscript.

Kind regards,

Li Yang, M.D.

Academic Editor

PLOS ONE

Journal Requirements:

Reviewers' comments:

Reviewer's Responses to Questions

**Comments to the Author**

1. If the authors have adequately addressed your comments raised in a previous round of review and you feel that this manuscript is now acceptable for publication, you may indicate that here to bypass the “Comments to the Author” section, enter your conflict of interest statement in the “Confidential to Editor” section, and submit your "Accept" recommendation.

Reviewer #1: All comments have been addressed

Reviewer #4: All comments have been addressed

Reviewer #5: (No Response)

Reviewer #6: (No Response)

2. Is the manuscript technically sound, and do the data support the conclusions?

Reviewer #1: Yes

Reviewer #4: Yes

Reviewer #5: Yes

Reviewer #6: Yes

3. Has the statistical analysis been performed appropriately and rigorously? 

Reviewer #1: Yes

Reviewer #4: Yes

Reviewer #5: Yes

Reviewer #6: Yes

4. Have the authors made all data underlying the findings in their manuscript fully available?

Reviewer #1: Yes

Reviewer #4: Yes

Reviewer #5: Yes

Reviewer #6: Yes

5. Is the manuscript presented in an intelligible fashion and written in standard English?

Reviewer #1: Yes

Reviewer #4: Yes

Reviewer #5: Yes

Reviewer #6: Yes

6. Review Comments to the Author

Reviewer #1: The author has made meaningful revisions to the manuscript. The author has added background information to the study. The author has supplemented the progress of research on machine learning and disease prognosis. The limitations of the sample size and model have been objectively addressed. The addition of clinical significance is appropriate.

Reviewer #4: The author of this paper addressed all the mentioned concerns in their previous version.

Please accept this latest version.

No further comments from my side.

Thanks

Reviewer #5: The article was reviewed under the title "Advancing patient care: Machine learning models for predicting grade 3+ toxicities in gynecologic cancer patients treated with HDR brachytherapy".

Overall, the focus of the study is good. However, I would like to offer several recommendations that authors may find useful in the process of revising their manuscript:

1- Given the importance of the research topic, it is necessary to examine previous studies in more detail.

2- Accordingly, a comparison between the results obtained and other research can be described.

3- Considering the recent research conducted on gynecological diseases based on deep learning methods, is there a specific reason for choosing classical machine learning methods?

4- Some of the sources used are old, and given the more recent research, it is better to use newer sources for reference.

5- Given the need for large training data for machine learning models, is there a reason to use small amounts of data?

Reviewer #6: The manuscript introduces the results of a study that built and compared some of the more common binary classification machine learning models in the context of predicting if a patient is going to develop grade 3 or higher toxicities in gynecologic cancer patients treated with EBRT and brachytherapy. While the motivation and the relevance of the work is high, the approaches discussed not necessarily the most performant. The task itself is not a very complex one (i.e., binary classification) with a note that 3+ toxicities are not a large sample. Overall, the work has its merits through the thorough and systematic approach to evaluate and compare the results among the ML models.

I would say, that the manuscript resembles a lab report rather than a journal manuscript. I have added more technical comments and suggestions below.

- you mention on line 130 efficiency and reproducibility. I salute the code release but I would be careful on the "efficiency" statement. What does that mean in your case. Efficient implementations are typically linked to the underlying resources where the model are trained and used in inference. There we have different metrics for performance. Without values (training time, resources used), I would refrain to talk about efficiency.

- on line 135 you mention imputation as a second step in your pipeline. Although you mention SOTA methods later, it is unclear why would you need it. Some of the models there also do well with little data, and given that the 3+ toxicity are relatively rare (see line 154 and 155) w.r.t. the total treated patients, this might need some motivation. I would argue you could do well enough with "good priors" and a simpler model.

- I would suggest to also look at XGBoost as a candidate ML model for better performance as the out of the box SKLearn SVM and Random forest, multiple advantages event when considering efficiency (the optimized XGBoost model can run on a PC, tablet, even in the browser)

- the evaluation metrics are standard, I would move them in appendix, at least for the ML community it is redundant

- the model choice is "a la textbook", rather than supported by intuitive insights on what, how, and why each model can provide for the task at hand

- the limitations (line 432) is not necessarily a limitation, is the task itself. Again, chosing model which can do well on small datasets is the key (here statistics and simpler models can do way better than MLP)

- How are you planning to use the results to close the loop for the benefit of the patients in therapy control/management and QoL? Any thoughts about deployment, turning this in an app for the physicians?

- Did you use physicians when considering priors one can inject in the ML models to make sure they extract the correct dependency between variables? -

- Physics-informed ML or PhyML is a strong candidate to introduce and induce known (mechanistic priors) into the system and when complemented with simulation you can go beyond the statistical imputation of data

7. PLOS authors have the option to publish the peer review history of their article (what does this mean? ). If published, this will include your full peer review and any attached files.

**Do you want your identity to be public for this peer review?** For information about this choice, including consent withdrawal, please see our Privacy Policy .

Reviewer #1: No

Reviewer #4: No

Reviewer #5: No

Reviewer #6: No

---

## [Author Response · Author response to Decision Letter 1]

3 Apr 2025

Academic Editor

PLOS ONE

Dear Dr. Li Yang;

Title: Advancing Patient Care: Machine Learning Models for Predicting Grade 3+ Toxicities in Gynecologic Cancer Patients treated with HDR Brachytherapy

Thank you for your valuable suggestions and remarks from the editor and reviewers in this second round. We are very pleased to hear we addressed all the initial reviewer comments. After incorporating this new feedback, we feel our manuscript has greatly improved. Please find below the point-by-point responses to your comments. We hope that the present version of the manuscript will be acceptable for publication, and we look forward to hearing from you.

---

## [Decision Letter · Decision Letter 2]

16 Apr 2025

Advancing patient care: Machine learning models for predicting grade 3+ toxicities in gynecologic cancer patients treated with HDR brachytherapy

PONE-D-24-43669R2

Dear Dr. Portocarrero Bonifaz,

We’re pleased to inform you that your manuscript has been judged scientifically suitable for publication and will be formally accepted for publication once it meets all outstanding technical requirements.

Kind regards,

Li Yang, M.D.

Academic Editor

PLOS ONE

Additional Editor Comments (optional):

Thanks for the authors' efforts to comprehensively improve your manuscript according to editor's and reviewers' comments. I am pleased to inform you that your paper can be accepted for publication now. Thanks for the chance to assess your interesting and important work. Additionally, many thanks for all the reviewers' precious inputs.

Reviewers' comments:

Reviewer's Responses to Questions

**Comments to the Author**

1. If the authors have adequately addressed your comments raised in a previous round of review and you feel that this manuscript is now acceptable for publication, you may indicate that here to bypass the “Comments to the Author” section, enter your conflict of interest statement in the “Confidential to Editor” section, and submit your "Accept" recommendation.

Reviewer #5: (No Response)

2. Is the manuscript technically sound, and do the data support the conclusions?

Reviewer #5: (No Response)

3. Has the statistical analysis been performed appropriately and rigorously? 

Reviewer #5: (No Response)

4. Have the authors made all data underlying the findings in their manuscript fully available?

Reviewer #5: (No Response)

5. Is the manuscript presented in an intelligible fashion and written in standard English?

Reviewer #5: (No Response)

6. Review Comments to the Author

Reviewer #5: (No Response)

7. PLOS authors have the option to publish the peer review history of their article (what does this mean? ). If published, this will include your full peer review and any attached files.

**Do you want your identity to be public for this peer review?** For information about this choice, including consent withdrawal, please see our Privacy Policy .

Reviewer #5: No

---

## [Editor Report · Acceptance letter]

PONE-D-24-43669R2

PLOS ONE

Dear Dr. Portocarrero Bonifaz,

I'm pleased to inform you that your manuscript has been deemed suitable for publication in PLOS ONE. Congratulations! Your manuscript is now being handed over to our production team.

Kind regards,

on behalf of

Dr. Li Yang

Academic Editor

PLOS ONE